# Potential Role of Vitamins A, B, C, D and E in TB Treatment and Prevention: A Narrative Review

**DOI:** 10.3390/antibiotics10111354

**Published:** 2021-11-05

**Authors:** Giulia Patti, Carmen Pellegrino, Aurelia Ricciardi, Roberta Novara, Sergio Cotugno, Roberta Papagni, Giacomo Guido, Valentina Totaro, Giuseppina De Iaco, Federica Romanelli, Stefania Stolfa, Maria Letizia Minardi, Luigi Ronga, Ilenia Fato, Rossana Lattanzio, Davide Fiore Bavaro, Gina Gualano, Loredana Sarmati, Annalisa Saracino, Fabrizio Palmieri, Francesco Di Gennaro

**Affiliations:** 1Clinic of Infectious Diseases, Department of Biomedical Sciences and Human Oncology, University of Bari “Aldo Moro”, 70123 Bari, Italy; giuliapatti22@gmail.com (G.P.); karmenpellegrino@gmail.com (C.P.); aurelia.r.92@gmail.com (A.R.); robertanovara@gmail.com (R.N.); sergio.cotugno@gmail.com (S.C.); robertapapagni0@gmail.com (R.P.); giacguido@gmail.com (G.G.); valenduzza@hotmail.it (V.T.); giusideiaco@gmail.com (G.D.I.); rossana.lattanzio@gmail.com (R.L.); davidebavaro@gmail.com (D.F.B.); annalisa.saracino@uniba.it (A.S.); cicciodigennaro@yahoo.it (F.D.G.); 2Microbiology and Virology Unit, University of Bari, University Hospital Policlinico, 70124 Bari, Italy; federicarosarosaromanelli@gmail.com (F.R.); stolfastefania@gmail.com (S.S.); rongalu@yahoo.it (L.R.); 3Infectious Diseases Clinic, University Hospital “Tor Vergata”, Department of Systems Medicine, University of Rome Tor Vergata, 00173 Rome, Italy; letiziaminardi@gmail.com (M.L.M.); Ilenia.fato@inmi.it (I.F.); sarmati@med.uniroma2.it (L.S.); 4National Institute for Infectious Diseases “L. Spallanzani” IRCCS, 00161 Rome, Italy; Fabrizio.palmieri@inmi.it

**Keywords:** tuberculosis, vitamin A, vitamin B, vitamin C, vitamin D, vitamin E, *M*. *tuberculosis*

## Abstract

(1) Background: Tuberculosis (TB) is one of the world’s top infectious killers, in fact every year 10 million people fall ill with TB and 1.5 million people die from TB. Vitamins have an important role in vital functions, due to their anti-oxidant, pro-oxidant, anti-inflammatory effects and to metabolic functions. The aim of this review is to discuss and summarize the evidence and still open questions regarding vitamin supplementation as a prophylactic measure in those who are at high risk of *Mycobacterium tuberculosis* (MTB) infection and active TB; (2) Methods: We conducted a search on PubMed, Scopus, Google Scholar, EMBASE, Cochrane Library and WHO websites starting from March 1950 to September 2021, in order to identify articles discussing the role of Vitamins A, B, C, D and E and Tuberculosis; (3) Results: Supplementation with multiple micronutrients (including zinc) rather than vitamin A alone may be more beneficial in TB. The WHO recommend Pyridoxine (vitamin B6) when high-dose isoniazid is administered. High concentrations of vitamin C sterilize drug-susceptible, MDR and extensively drug-resistant MTB cultures and prevent the emergence of drug persisters; Vitamin D suppresses the replication of mycobacterium in vitro while VE showed a promising role in TB management as a result of its connection with oxidative balance; (4) Conclusions: Our review suggests and encourages the use of vitamins in TB patients. In fact, their use may improve outcomes by helping both nutritionally and by interacting directly and/or indirectly with MTB. Several and more comprehensive trials are needed to reinforce these suggestions.

## 1. Introduction

Tuberculosis (TB) is one of the world’s top infectious killers. In fact every year 10 million people fall ill with TB and 1.5 million people die from TB [1]; this is just the tip of the iceberg. The incidence of TB is higher in low-income settings where people are more likely to be malnourished, resulting in lower systemic levels of immunomodulatory vitamins. Malnutrition and undernutrition have long been recognized as major risk factors for all infectious diseases, especially tuberculosis [2,3]. Vitamins are organic bio-molecules that must be obtained from food, except vit D, which can be synthetized in the human body after exposure to sunlight. Vitamins have an important role in vital functions, due to their anti-oxidant, pro-oxidant, anti-inflammatory effects, and to metabolic functions [4]; their role is important both in the latent tuberculosis infection (LTBI) and in the active disease and for all ages of tb patients [4,5]. The anti-tubercular effects of vitamins have been studied since the pre-antibiotic era, but until today there is no definitive evidence for vitamin supplementation to treat the disease or the infection. The aim of this review is to discuss and summarize the evidence and still open questions regarding vitamin supplementation as a prophylactic measure in those who are at high risk of *Mycobacterium tuberculosis* (MTB) infection and active TB.

## 2. Results

### 2.1. Vitamin A

Vitamin A (VA) is a lipid-soluble molecule that belongs to the retinoic acid family. It is made up of four isoprenoid units linked head to tail. Provitamin A carotenoids (beta-carotene, alpha-carotene, and beta-cryptoxanthin) and preformed vitamin A are the two major forms of vitamin A [4]. Table 1 summerize main findigs on VA and Tuberculosis infection.

Various studies found association between VA deficiency and a variety of infectious diseases, including diarrheal and respiratory diseases, schistosomiasis, malaria, tuberculosis, leprosy, rheumatic fever, and otitis media [6,7]. Vitamin A is involved in both T- and B-lymphocyte function, the generation of antibody responses, and macrophage activity [8]. Several studies demonstrated that VA and its metabolites play a role in the pathogenesis of TB. While low levels of the circulating form of VA, retinol, correlate with susceptibility to disease with a 10-fold increased risk of TB [9,10], high levels of the bioactive hormonal form of vitamin A, all-trans retinoic acid (ATRA), induces antimicrobial activity in *M. tuberculosis*-infected macrophages, as it has been shown in laboratory studies [11].

Many researchers have looked at the effects of VA supplementation in patients with pulmonary tuberculosis, and many of them have looked at vitamin A and zinc co-administration. Karyadi et al. [12] found that vitamin A and zinc supplementation improves the impact of tuberculosis medicine after two months of antituberculosis treatment and resulted in earlier sputum smear conversion in a double-blind, placebo-controlled study [12]. On the other hand, a randomized controlled trial demonstrated that supplementation with vitamin A and zinc did not affect treatment outcomes at eight weeks [13].

Another study found no improvement in sputum smear conversion with zinc and vitamin A supplementation [14]; Hanekom et al. [15] did not observe any effect of high dose vitamin A therapy on disease outcomes in South African children with TB [15]; a Tanzanian trial reported no effect of vitamin A and other vitamin and mineral supplementation on sputum culture conversion and overall survival in patients with pulmonary TB [16,17]. In a review of vitamin A and zinc Christian and West [18] concluded that zinc deficiency could cause a secondary VA deficiency in protein–energy deficient populations and disease states with severely compromised liver function. Zinc is in fact required for the mobilization of VA from the liver, making it a key component in vitamin A metabolism [18]. This may be the explanation for a strong association between vitamin A deficiency and wasting observed in patients with TB [19,20]. Therefore, supplementation with VA may not be beneficial in TB without correction of the zinc deficiency.

In a small study, Chandra showed better sputum conversion with antituberculosis treatment (ATT) supplemented with multivitamin–trace elements [21]. Range et al. [16,17] in their clinical study, as already mentioned, tested the supplementation of multiple micronutrients, including vitamin A but not zinc, without positive results on the rate of sputum culture conversion.

A Peruvian study found that VA deficiency strongly predicted risk of incident TB disease among household contacts of TB patients. These data suggest that vitamin A supplementation among individuals at high risk of TB may provide an effective means of preventing TB disease [9].

Furthermore, low VA levels are common not only in patients with active tuberculosis but also in HIV patients, and are even more severe among patients with HIV/TB co-infection [20,22]. Mugusi et al. [22] observed a highly significant improvement in levels of vitamin A in HIV-negative tuberculosis patients at two months from treatment [22]. Few studies have shown that low vitamin A levels return to normal after ATT even in the absence of vitamin A supplementation [10,22,23]. Specifically, it is difficult to know whether the vitamin A deficiency led to TB, or whether TB led to the vitamin A deficiency. Malnutrition predisposes to TB and TB causes ‘consumption’ [21,22]. Vitamin A and zinc deficiency reduces host defenses and immune responses but clear evidence is lacking of the benefits of VA supplementation in TB. Zinc deficiency could both precipitate immune system conditions and lead to a secondary vitamin A deficiency [9,22].

### 2.2. Vitamin B Complex

The vitamin B-complex vitamins include: B1 (thiamine), B2 (riboflavin), B3 (niacin), B5 (pantothenic acid), B6 (pyridoxine), B7 (biotin), B9 (folic acid), and B12 (cobalamins).

#### 2.2.1. Vitamin B1 (Thiamine)

VB1 promotes macrophage polarization into classically activated phenotypes with strong microbicidal activity and increased tumor necrosis factor (TNF-) and interleukin-6 expression, at least in part by promoting nuclear factor-B signaling, by modulating peroxisome proliferator-activated receptor (PPAR-). VB1 also increases mitochondrial respiration and lipid metabolism, whereas PPAR- integrates (PPAR-) metabolic and inflammatory signaling controlled by VB1. Hu S. et al. found that macrophages from VB1-treated mice infected with MTB have increased expression levels of CD86 and MHC-II, which are characteristic features of classically activated macrophages [24]. TNF-α, TNF-, IL-6, and nitrate levels were found to be higher in the lungs of VB1-treated animals [24]. Furthermore thiamin, in its active state thiamin diphosphate, is an important vitamin required for amino acid and carbohydrate metabolism enzymes. In addition, MTB lacks thiamin salvage mechanisms, making thiamine biosynthetic processes appealing therapeutic targets.

#### 2.2.2. Vitamin B2 (Riboflavin)

Usage of riboflavin as part of antitubercular treatment has been explored since 1962 by Huempfner R, who showed no adding value of riboflavin added to isoniazid. In a recent paper, the authors supposed a lower lifetime risk of developing active tuberculosis by activation of mucosal-associated invariant T cells (MAIT) from riboflavin precursor metabolites, increasing protection in the early stage of mycobacterial infections [25,26].

Bhaiyyasaheb Harale et al., showed that vitamin B2 (VB2) is indispensable for flavoenzymes, cofactors in redox reactions. Flavoproteins take part in a variety of redox reactions, catalyze oxidization and hydroxylation. In fact, flavin mononucleotide (FMN) riboswitch binded to synthetic riboflavin analogs may repress the adjacent riboflavin synthetic genes expression, and this could have resulted in inhibition of *M. tuberculosis* growth. This could be dependent upon the endogenous biosynthesis of riboflavin since *M. tuberculosis* lacks riboflavin transporter [27]. VB2 appears to participate in early response to MTB infections, activating innate T-cells. In addition, it is clear that the thiamin biosynthetic pathway can be targeted by drugs. Studying mycobacterial genes from the perspective of new drug targets is therefore imperative, and pathways leading to the biosynthesis of biotin, thiamine and pyridoxine form interesting study areas [27]. The role of Vitamin B is also relevant during tuberculosis treatment, in fact Pyrazinamide and isoniazid frequently results in vitamin-B6 insufficiency, which leads to a variety of peripheral neuropathies [28].

#### 2.2.3. Vitamin B5 (Pantothenic acid)

Wenting He et al. investigated the influence of VB5 on inflammatory signaling pathways in macrophages infected with MTB. The VB5 enhanced phosphorylation of nuclear factor-κB (NF-κB), Protein kinase B (PKB), also known as Akt, and p38, while suppressing the early phosphorylation of ERK. Levels of both TNF-α and IL-6 protein were significantly higher in the VB5-treated BMDMs (one marrow-derived macrophages (BMDMs). However, there was no difference in IL-4, IL-10, or IL-13 protein levels between the VB5-treated BMDMs and the control group. Overall, their findings showed that after mycobacterial infection, VB5 mostly had a proinflammatory effect in macrophages [29].

#### 2.2.4. Vitamin B6 (Pyridoxine)

Vitamin B6 is well-known for its antioxidant properties, which include the capacity to scavenge reactive oxygen species (ROS) [30]. A heteromeric PLP synthase comprised of Pdx1 and Pdx2 was found to produce pyridoxal 5-phosphate (PLP), the bioactive form of vitamin B6, in MTB. A B6 auxotrophic MTB mutant is produced when the pdx1 gene is damaged. By eliminating the cofactor during exponential growth or stationary phase, the necessity of vitamin B6 synthesis for pathogen growth and survival in culture is proven [31].

Vitamin B6 (pyridoxine) supplementation during isoniazid (INH) therapy is necessary to prevent the development of peripheral neuropathy [28,32].

#### 2.2.5. Vitamin B7 (Biotin)

*M. tuberculosis* requires vitamin B7 (biotin). It functions as a cofactor in two essential enzymes involved in fatty acid synthesis and anaplerosis: acyl CoA carboxylase and pyruvate carboxylase. *M. tuberculosis* is said to need de novo biotin synthesis since it lacks biotin transporters, as evidenced by genetic research [33]. The biotin concentration in the human host’s serum is insufficient to satisfy the microorganism’s needs. Biotin is made by the enzymes BioF, BioA, BioD, and BioB, which need pimeloyl-CoA as a precursor. BioF, bioA, and bioB are essential for *M. tuberculosis* virulence and pathogenicity, according to Sassetti et al., who discovered this using genome-wide genetic screens [33]. To establish biotin biosynthesis genes as prime targets for antituberculous action, more study is needed [34], but Wanisa Salaemae et al. found that amiclenomycin, a BioA inhibitor, has an activity against Mycobacteria spp., only species that can scavenge exogenous biotin [35]. Consequently, *M. tuberculosis* needs biotin production so that we can observe enzymes that synthetize biotin as promising drug targets for new antibiotics [36].

The vitamin B7 (required by *M. tuberculosis* for development and pathogenicity) synthesis genes could be therefore important targets of antituberculous therapy, as biotin is an important substrate for MTB growth and virulence [33,36].

#### 2.2.6. Vitamin B12 (Cobalamin)

*M. tuberculosis* is one of a few number of microorganisms possessing the capacity for de novo biosynthesis of vitamin B12 (VB12). VB12 and related corrinoids can also be scavenged by the bacillus using an ATP-binding cassette [37]. While the modest numbers of tuberculosis bacteria are unlikely to have an effect on overall VB12 levels in infected people, vitamin B12 generated or ingested by mycobacterial pathogens may have a local effect on the metabolic environment of human granulomatous lesions. As highlighted by Boritsch et al. [38], the cobF gene (encoding a precorrin-6a-synthase required for B12 synthesi) was lost during evolution from the M. canettii-like ancestor. Despite these deletions, B12 biosynthesis was not entirely depleted [39]. The MTB regulates its core metabolic functions according to B12 availability, through the acquisition of B12 via endogenous or through uptake from the environment. A role of vitamin B12 in pathogenesis may be implicated, but this finding is still poorly understood [37]. In a previous study by Chanarin et al., the incidence of tuberculosis in vegetarians was 133 per 1000, while it was 48 per 1000 in patients on varied diets. There is a second intriguing hypothesis that may be relevant: in cobalamin deficiency there is a failure to convert methylmalonic acid to succinic acid as cobalamin is a coenzyme in the enzyme methylmalonyl CoA-mutase. One of the components of the cell wall of *M. tuberculosis* is mycocerosic acid, and methylmalonic acid is the obligatory substrate for its synthesis. The increased concentrations of methylmalonic acid present in vitamin B12-deficient vegetarians could provide a particularly favorable climate for the multiplication of *M. tuberculosis*.

Vitamin B12 concentration in the blood could be a biomarker in the identification of active tuberculosis with and without HIV infection, according to a recent study. Detecting the concentration level of these micronutrients in HIV-uninfected patients could be used as a TB treatment response monitoring biomarker [40].

In addiction, authors showed potent antimycobacterial activity with a minimum inhibitory concentration (MIC) of 6 μg/mL [33]. Furthermore, Vitamin B12 insufficiency has also been linked to neuropathy in patients with ileal TB [41].

### 2.3. Vitamin C

In humans, vitamin C (VC) is an essential dietary micronutrient with antioxidant and prooxidant property which is involved in a wide range of vital cellular and physiological functions [42].

In several reports, authors described an improved effect of orally administered VC in preventing and treating *M. tuberculosis* infection in humans and animals [43,44]; others have reported that the malnutrition and the deficiency of VC in TB patients is related to a high risk of developing the disease and also developing severe outcomes [45,46,47].

Different studies have focused on understanding how *M. tuberculosis* interacts with VC, with controversial findings.

Recently Vilchèze et al. discovered that VC sterilizes cultures of drug susceptible and drug resistant *M. tuberculosis* by induced Fenton reaction [48]. In this study, authors demonstrated that the ability of VC to sterilize *M. tuberculosis* cultures results from an increase in ferrous ion concentration leading to ROS production, lipid alterations, redox unbalance and DNA damage [48].

The sterilizing effect of VC was more substantial in Tuberculosis strains deficient for mycothiol [49,50]. Yet, the combination of a mycothiol inhibitor and VC or other pro-oxidant compound could lead to more rapid *M. tuberculosis* cell death than with most antibiotics used [48,49]. VC has also an effect on lipid biosynthesis. In VC treated-MTB cultures the authors observed the generation of two free 2-hydroxylated long-chain fatty acids [48]. They compared their findings to those of Kondo e Kanai [51], who found that 2-hydroxylated fatty acids were more toxic to mycobacteria than the equivalent saturated fatty acids. Vichèze and colleagues describe that the accumulation of these fatty acids could lead to a bactericidal event in tuberculosis [48]. Furthermore, the reduction in phospholipid content observed in VC-treated *M. tuberculosis* could affect the mycobacterial cell wall structure and, as a result, *M. tuberculosis* survival [48]. In addition, VC is believed to reduce the level of guanosine 5′-diphosphate 3′-diphosphate (ppGpp), a molecule thought to be involved in growth regulation and stress response in *M. tuberculosis* [52,53]. The VC potentiates the Pyrazinamide (PZA) action, an essential first line TB drug because of its sterilizing activity against non-metabolizing/slowly metabolizing persistent bacilli that are resistant to other drugs, by tackling dormant organisms and inhibiting the development of rifampin-tolerant and rifampin-resistant bacilli [52]. Anyway these studies reached divergent conclusions [54,55]. Vilchèze, Hartman et al., demonstrated that high concentrations of VC sterilize drug-susceptible, and drug-resistant tuberculosis cultures and prevent the emergence of drug persisters, discovered in the same study, and that the combination of a sub-lethal concentration of VC with first line antitubercular drug Isoniazid (INH) resulted in sterilization of *M. tuberculosis* cultures after 4 weeks, with no resistant mutant escape in the treatment. This suggests that the addition of a sub-inhibitory dose of VC to INH treatment could also reduce the emergence of INH-resistant mutants [48]. In another study VC alone had no activity against *M. tuberculosis* in mice but had the potential to boost the efficacy of INH-RIF treatment, thus increasing the elimination rate of *M. tuberculosis* in mice compared to that with INH-RIF alone [56].

Low levels of micronutrients are commonly observed among TB patients [57,58]. Lower blood levels of vitamins A and C have been reported to be associated with higher TB incidence [59]. Charpy et al. observed that giving terminally ill TB patients daily high dosages (15 g/day) of VC orally for 6 to 8 months enhanced appetite and physical activity with no negative effects [60]. Although no radiological improvement in the disease was observed, the authors concluded that the addition of VC to TB treatment might improve patients’ resistance to infection. As Vilchèze et al. have shown in their studies in vitro, VC activity was dependent on the iron concentration present in the media. When a low concentration of ferric ion was present or when the iron chelator deferoxamine was used, VC had no activity in vitro [48]. Nramp transports iron and regulates iron levels within the phagosome of macrophages where *M. tuberculosis* resides [61]. Human Nramp1 polymorphisms, which have been associated with TB resistance [62], as well as polymorphisms found in the VC transporters encoded by SLC23A1 and SLC23A2 [63], may indicate that the host genotype could impact the efficacy of VC as an adjunct therapy [64,65].

### 2.4. Vitamin D

Vitamin D (VD) may play a fundamental role in the innate and adaptive immune response against MTB infection and in the progression of the disease. MTB enters the body through the TLRs present on macrophages. Activation of the signaling pathway of TLR and exposure to inflammatory cytokines induce the expression of CYP27B1 oxidase by macrophages. This oxidase is the one that converts 25(OH)-VD into the active form 1,25(OH)-VD, which, in autocrine mode, stimulates the antimicrobial activity of macrophages through the VDR/RXR signaling pathway, inducing the endogenous production of cathelicidin hCAP-18 and its derivative antimicrobial peptide LL-37. LL37 is able to interact with the molecules of the bacterial wall and perforate the cytoplasmic membrane, resulting in the death of the bacterial cell [66,67,68,69].

It also seems that VD induces autophagy in infected macrophages [70]. Autophagy is a process that induces the maturation of phagolysosomes and the degradation of intracellular pathogens. While MTB tries to block the autophagy process, MTB lipoproteins stimulate it by activating the signaling pathway mediated by TLR2/1, CD14 and VDR [71]. In particular, it has been noted that by silencing the expression of LL-37, autophagy is blocked, allowing us to suppose that this antibacterial mechanism depends precisely on the cathelicidin/LL-37 system [72]. In another study it was shown that oral administration of 5000 IU of VD and 500 mg b.d. phenylbutyrate consistently increases the production of LL-37 in both macrophages and lymphocytes and leads to an increase in intracellular killing of MTB by macrophages [73].

VD is also required for acquired immunity to operate against MTB infection. Th1 cells secrete INF γ, which causes macrophages to produce antimicrobial compounds, as well as autophagy, phagosome maturation, and other anti-MTB effects. Th2 lymphocytes induce an antibody-mediated response through the secretion of IL-4 and IL-5. It seems that these antimycobacterial activities do not occur in case of VD deficiency [74,75].

Several studies highlight the anti-inflammatory role of VD, which determines population expansion of T-reg lymphocytes that inhibit the Th1 immune response and reduces the production of MMPs [76,77]. Furthermore, it appears that VD and its hydroxylated derivatives also promote the stabilization of the endothelium and of the barrier function in the presence of inflammatory mediators [78]. This is essential to reduce the inflammatory state and tissue damage that characterize the pathophysiology of tuberculous disease [78].

Several studies have shown that TB patients have low levels of 25(OH)-VD compared to healthy controls [79,80,81]. However, it is unclear whether VD deficiency is a possible cause or consequence of the disease. Numerous in vitro studies have demonstrated the role and potential benefit of VD in TB, showing that VD suppresses the replication of MTB in vitro [82,83]; few clinical trials have been performed to evaluate the efficacy of VD supplementation in preventing the development of the disease, failing to show a reduction in the risk of infection and development of disease [84,85].

However, in a meta-analysis of prospective studies, Aibana O, Huang C-C et al. found a positive dose-dependent correlation between low serum levels of 25(OH)-VD and the risk of developing tuberculous disease [86].

In another meta-analysis, VD deficiency was positively and significantly associated with an increased risk of TB and an increased risk of developing active TB in LTBI subjects/household contacts of active TB patients [80]. Due to heterogeneity in the definition of VD deficiency, which can affect the results of the meta-analysis, more studies are needed to evaluate the effectiveness of the VD supplement in the prevention of TB.

The use of VD alongside TB drugs is controversial: a study has demonstrated the efficacy of its use together with PZA in vitro [87,88], confirmed by an in vivo study conducted in mice by Jing Zhang et al. [89]. Simultaneous use of calcitriol and PZA leads to a faster reduction of lung lesions and a halt in bacterial growth and increase both in anti-inflammatory cytokines (i.e., IL-4), and in antimicrobial molecules such as cathelicidin/LL-37, which are reduced in patients where PZA is administered alone [89].

It has not yet been clarified whether the administration of the standard therapeutic regimen can affect VD levels in blood. In fact, the presence of liver damage can decrease the levels of the enzymes of activation by hydroxylation of VD; some studies found that the cytochrome P450 system is inhibited or induced by INH, resulting in an alteration of 25-hydroxylase and 1-hydroxylase [90,91]; furthermore, RIF can increase CYP3A4-dependent catabolism, but not CYP24A1, that is 25(OH)-VD-24-hydroxylase, thus reducing the concentration of active VD in the bloodstream [92,93]. There is no evidence on the impact of Ethambutol or PZA on the metabolism of calcitriol.

### 2.5. Vitamin E

Vitamin E (VE) has a lengthy isoprenoid side chain attached at the 2 position of a 6-chromanol ring, and this distinguishes vitamin E from other fat-soluble vitamins, conferring peculiar biochemical properties: it is a powerful antioxidant, it promotes membrane stability, and it inhibits platelet aggregation. Vitamin E has been shown to provide numerous health benefits, including protection against infections. Vitamin E has immunological qualities that include boosting the body’s defenses, improving humoral and cell immune responses, and increasing phagocytic functions [94]. VE is a potent antioxidant that removes free radicals and protects the structural components of cell membranes from peroxidation [95]), and it’s also involved in T-cell differentiation and proliferation, with a role in preventing bacterial and viral infections.

When studying the link between VE and TB, a controversial pathophysiological component should be considered: on the one hand, phagocytic immunity is a critical stage in the host response to TB; on the other hand, oxidative stress is a major reaction against *M. tuberculosis*. The trigger of this counteractive response is concentration-dependent: *M. tuberculosis* releases DNA damage-responsive gene only if ROS levels are low, thus this mechanism does not result in protection against high levels of oxidative stress [96].

It has been observed that the nutritional status of TB patients shows lower levels of VE than non-TB people (together with other antioxidant agents such as VA, anti-oxidase and beta-carotene) [97]. The level of VE has been analyzed at the starting time of antituberculosis treatment (ATT) and during the course of ATT: poor levels have been observed but no increase related to the assumption of ATT [98,99]. A study found negative connections between VE (and hence vitamin C and A) and oxidative stress, and nitrosative stress markers such as nitric oxide, carbonyl protein, and lipid peroxidation in TB [100]. In patients who recovered well, the alterations were reversed after six months of antitubercular medication, although the elevated stress was not totally removed [100]. VE has been claimed to play a part in boosting the level of immunoglobulins and T-cell types against TB when it comes to immunity empowerment [101].

From an epidemiological viewpoint, Aibana and colleagues have discovered a clinical match to laboratory findings of VE insufficiency and TB, observing that VE deficiency is linked to an elevated risk of TB progression in index TB case household contacts [102].

The in vivo link between VE and ATT has been studied by certain researchers. Pretreatment with high dosages of tocopherol (100 mg/kg) may protect albino rabbits from Isoniazid and Rifampicin-induced hepatotoxicity [103]. It was shown that VE (and VC) positively controlled the sperm quality, hematological, and brain damage caused by anti-tuberculous drugs in rats [104,105].

VE supplementation has also been investigated for its potential benefits. In this line, a research of 92 MDR-PTB patients conducted in Indonesia found that VE (as well as fat) intake improves quality of life [106]. Nonetheless, different study comes to no conclusions about this, and suggests that the only observed benefit from vitamin supplementation may be linked to weight growth rather than a direct good TB outcome and also on other infectious diseases such as surgical site infection and pneumonia [107]. In a recent study Groebler and colleagues have validated this suspicion, noting that there is insufficient evidence to recommend routine supplementation in TB patients, despite indications of reduced vitamin blood levels [108]. Furthermore, Hemilä H and colleagues, who conducted a prospective cohort research in Finland, found that VE may enhance the risk of TB in people who smoked heavily [109].

A Chinese study, on the other hand, has looked at vitamin supplementation from the perspective of public health. Ren Z and colleagues published a cross-sectional study where they found an insufficient level of macronutrients and micronutrients such as VTE in TB patients, particularly among the unemployed and those eating at home. As a result, the authors advocate for public health action to focus on food policy for TB patients [110].

Finally, the contribution of Srinivasan S and colleagues in the field of urogenital TB deserves to be acknowledged. VE supplementation, according to two studies published in 2004, lowers the incidence of kidney stone formation by preserving the membrane [111,112].

## 3. Materials and Methods

We conducted a search on PubMed, Scopus, Google Scholar, EMBASE, Cochrane Library and WHO websites (http://www.who.int, accessed on 26 September 2021) starting from March 1950 to September 2021, in order to identify articles discussing the role of Vitamins A, B, C, D and E and TB. We included all articles addressing epidemiology, physiopathology, clinical features, screening and diagnosis, treatment and management.

## 4. Conclusions

Our reading review shows a strong link between vitamins A, B, C, D and E and TB. This seems to be due both to their modulatory action in inflammation process, to the nutritional status which, where it is poor, correlates on the one hand with a hypovitaminosis and on the other with an increased risk of TB onset, and to the action of vitamins which potentially have a direct link with *M. tuberculosis.*

Supplementation with multiple micronutrients (including zinc) rather than VA alone may be more beneficial in TB. In the absence of evidence of the beneficial effect of VA supplementation alone, interventions that include both VA and zinc may be a better choice [38].

WHO recommend Pyridoxine (vitamin B6) when high-dose Isoniazid is administered and in patients with diabetes, uremia, HIV infection, seizure disorders, alcohol abuse, malnutrition or peripheral neuropathy. Additionally, pregnant and postpartum women and exclusively breastfed infants should receive vitamin B6 while taking Isoniazid. (Normal dose of pyridoxine when used prophylactically for prevention of neuropathy in patients taking Isoniazid is 10–25 mg/day.)

For VC, the daily dose in a normal human diet is far below the levels needed for optimal activity in vitro against MTB; megadoses of VC have been administered to human patients, leading to high VC plasma concentrations (up to 49 mM), which were shown to be safe [64]. So it will be very interesting performing randomized clinical trials in the future prospect to better understand if VC is a means to achieving faster sputum smear-negative conversion, which would indicate a quicker reduction in the mycobacterial burden, and if it might shorten TB treatment, leading to a reduction in the incidences of both drug resistance development and disease relapse.

While for VD, considering the few side effects, even if administered in high doses, given the low cost of the drug and given the evidence of in vitro efficacy and dubious in vivo results, it would be useful to measure VD in all patients with TB, and in patients at risk of developing active TB, and administer supplementation of VD. However, further clinical trials are necessary to identify any confounding factors in vivo and to determine the possible usefulness of VD in the prophylaxis and treatment of TB, and to determine the dosages and the most suitable method of administration to obtain benefit.

In addition, VE showed a promising role in TB management as a result of its connection with oxidative balance. This statement is supported by laboratories findings. Hence, VE supplementation likely protects against TB progression in household contacts, against ATT-related organ toxicity and indirect complication of extrapulmonary TB such as kidney stone formation. Nevertheless, VE intake in excess could have adverse effects (as any vitamin), so it should not be subministrated indefinitely. More studies are required to profile which patients would benefit at best. No studies focus on positive sputum time-conversion.

Our review suggests and encourages the use of vitamins in TB patients. In fact, their use may improve outcomes by helping both nutritionally and by interacting directly and/or indirectly with MTB. Several and more comprehensive trials are needed to reinforce these suggestions.

## Figures and Tables

**Table 1 antibiotics-10-01354-t001:** Summary of Impact of Vitamins A, B, C, D and E on *M. tuberculosis* infection.

Vitamin	Type of Vitamin	Physiopathology	Potential Role	Trial (Yes/No)	Results
A	Lipid-soluble	-Cellular differentiation -Integrity of the mucosal epithelia -Increased immune system responses (lymphocytes proliferation, macrophage activity, switch type 2/type 1 cytokines)	-Vitamin A deficiency predicted risk of incident TB disease in people exposed to *M. tuberculosis.* -No improvement in sputum smear conversion with vitamin A supplementation-Better sputum conversion with multivitamin–trace elements supplementation	[13,14,15,16,17,27]	Vitamin A did not significantly affect the time to smear conversion in patients with pulmonary TB
B1	Hydrosoluble	Promotes macrophage polarization into classically activated phenotypes with strong microbicidal activity and enhanced tumor necrosis factor-α and interleukin-6 expression, at least in part by promoting nuclear factor-κB signaling	Improve the protective immune response to reduce MTB survival inside macrophages and in vivo VB1	[24]	Improve the mice’s protective immunological response during MTB infection.
B2	Hydrosoluble	It is indispensable for flavoenzymes, cofactors in redox reactions	Improve innate T cell response in early MTB infection	[25,27]	VB2 may protect on early stages of MTB infections
B5	Hydrosoluble	VB5 enhanced the phosphorylation of nuclear factor-κB (NF-κB), Protein kinase B (PKB), also known as Akt, and p38, while suppressing the early phosphorylation of ERK. TNF- and IL-6 protein levels were significantly higher in the VB5-treated BMDMs	Macrophage clearance of intracellular mycobacteria	[29]	No differences between group control and VB5 treated patients
B6	Hydrosoluble	antioxidant properties	Scavenge ROS	[31]	Essential for growth and survival of *M. tuberculosis* in culture
B7	Hydrosoluble	*M. tuberculosis* requires vitamin B7 (biotin). It acts as a cofactor in acyl CoA carboxylase and pyruvate carboxylase, two important enzymes involved in fatty acid production and anaplerosis. It has been claimed that *M. tuberculosis* requires de novo biotin production since it lacks biotin transporters, as evidenced by genetic research.	Cofactor in acyl CoA carboxylase and pyruvate carboxylase, enzymes involved in fatty acid production and anaplerosis	[33]	Exogenous B7 is required for *M. tuberculosis* virulence and pathogenicity. Enzymes for biotin biosynthesis could be a target for future treatment
B12	Hydrosoluble	ATP-binding cassette-type protein of MTB participates in VB12 production	*M. tuberculosis* regulates its core metabolic functions according to B12 availability, whether the acquisition of B12 via endogenous or through uptake from the environment	[37,40]	CobF gene was one of the genes required in MTB production of VB12. Low B12 supplement provides a favorable situation for MTB multiplication
C	Water-soluble	-VC induced Fenton reaction in aerobic envinroment leading to ROS production, lipid alteration, redox unbalance and DNA demage in MTB coltures. -The combination treatment with VC and Mycothiol inhibitors could lead to more rapid MTB cells death. -VC affects lipid biosynthesis in MTB and induces a reduction in phospholipid content -Transcriptional adaptation, grow arrest and dormancy phenotype development is triggered by VC	-High concentrations of vitamin C sterilize drug-susceptible, MDR and extensively drug-resistant MTB cultures and prevent the emergence of drug persisters -Fenton reaction is the main mechanism of action of VC against MTB -VC potentiates the cidal action of PZA -Addition dose of VC to INH-RIF treatment could reduce the emergence of INH and RIF resistant mutants	-In vitro [48,52] -In mice [56] -Humans [59,60]	-In TB patients with high doses of VC added to first line treatment, there is no evidence of any regression of TB lesion. The effects on the bedridden patients were described as remarkable, they had regained appetite and physical activity -VC might improve patients resistance to infection
D	fat-soluble	-Stimulates endogenous production of hCAP-18/LL37 which interacts with the molecules of the bacterial wall and perforate the cytoplasmic membrane of the bacterial cell; -Induces autophagy in infected macrophages; -Determines expansion of T-reg lymphocytes that inhibit the Th1 immune response; -Reduces the production of MMPs, associated with tissue remodeling and the formation of tuberculous granulomas; -Promotes the stabilization of the endothelium and of the barrier function in the presence of inflammatory mediators	-VD suppresses the replication of mycobacterium in vitro; -Anti-tuberculous drugs reduce the production of cathelicidin hCAP18-LL37; -In patients receiving 4-drug therapy, despite VD supplementation, there is no significant increase in serum VD levels.	In vivo but few and with numerically small samples and data inaccuracies [52,53,73,84,85]	-VD and TB prevention: conflicting results that do not show a clear correlation in vivo between VD supplementation and prevention of disease development -VD and treatment of TB: infusion of high doses of VD did not reach an early microscopic negativization, compared to placebo, but in a metanalysis of RCT it seems that VD can accelerate sputum culture conversion in patients with multidrug-resistant pulmonary TB. -In patients receiving 4-drug therapy, despite VD supplementation, there is no significant increase in serum VD levels.
E	liposolubil	-Boosting humoral and cell immune responses, increasing phagocytic functions -Removal of free radicals -Protection of structural components of cell membranes against peroxidation	Low levels of VE on TB onset; docuemented poor assumption of VE in TB patients; protective role against TB progression in household contacts protection against organ toxicity of ATT in vivo studies; improvement of quality of life; improvement of nutritional status during ATT	[12,13]	No reliable evidence for routinary supplementation over the major outcomes (i.e., reduced death, increased cure rate at 6 and 12 months, improved nutritional status)

## Data Availability

The raw data are available from the corresponding author.

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
