# Peer review of "Potential Role of Vitamins A, B, C, D and E in TB Treatment and Prevention: A Narrative Review"

_antibiotics, 2021, doi:10.3390/antibiotics10111354_

Round 1

Reviewer 1 Report

The manuscript is a comprehensive review about vitamin intake and tuberculosis clinical outcomes. The research question is important, especially in consideration that the major burden of the disease weighs on countries were undernutrition and malnutrition are endemic. Therefore, in my view, the present work represents a useful tool for researchers and policymakers. However, the authors should address some minor comments:

Major comments

  • All article needs some English editing, and paragraph about vitamin B1 (lines 119-133) needs extensive English editing.
  • It may be considered to split section 5 into “Discussion” and a more concise “Conclusion”.

Minor comments

  • Whole manuscript: all microorganisms names should be written in italics, without capital letter on the species name: “ tuberculosis
  • Whole manuscript: when multiple references are inserted together, they should be written between the same parenthesis, as follows:[1, 2] or [1, 3-6].
  • Whole manuscript: references should be cited only by first author family name. All given names should be removed.

  • Line 45: “this is just the tip of the iceberg” needs reference.
  • Line 82: “A second Indonesian trial, found”: wrong syntax.
  • Line 87: “cross-sectional, observational, and supplementation clinical trials”: needs rewording.
  • Line 96-97: study 21 is cited twice in the same sentence.
  • Line 112: the term “consumption” needs definition and reference.
  • Line 127: “TNF-α TNF-” needs correction.
  • Line 136: “In a 136 recent paper” which one?
  • Line 142: bold characters should be removed
  • Lines 141-145: sentence is too long. Also, MTB should be written “ tuberculosis”.
  • Line 147: “Also it is clear that the thiamin biosynthetic pathway can be targeted by 147 drugs” Needs reference.
  • Lines 148-150: “Studying mycobacterial genes from the perspective of new drug targets is therefore imperative, and pathways leading to the biosynthesis of biotin, thiamine and pyridoxine form interesting study areas.” This sentence is more fitting, potentially, among the conclusions.
  • Line 154, 164, 274: The first time an acronym is used it should be placed between parentheses.
  • Line 164: “Reactive oxygen species (ROS). [30].” The dot should be removed.
  • Line 166: “The pdx1 gene is disrupted, resulting in a B6 auxotrophic MTB mutant.” This sentence needs explanation.
  • Line 173: “It acting as a cofactor in acyl CoA carboxylase and pyruvate carboxylase”: wrong syntax.
  • Line 176: “as evidenced by genetic research”: reference needed.
  • Lines 178 – 179: enzyme names are written both with and without capital letter.
  • Lines 184-85: “Consequently, M. Tuberculosis needs biotin production so we can watch at enzymes 184 that synthetize biotin as promising drug targets for new antibiotics” needs rewording.
  • Lines 186-88 are redundant.
  • Line 192: “ATP-binding cassette-type protein. [37]” wrong syntax. The same for lines 198, 201, 213, 220, 225, 234, 269, 286, 295, 301, 307, 314, 335.
  • Lines 234-238: [48] is cited multiple times in the same paragraph.
  • Line 238: “They compares these results with the study by Kondo e Kanai”: wrong syntax.
  • Line 262-264: “Charpy et al. revealed that terminally ill TB patients were given daily 262 high doses (15 g/day) of vitamin C orally for 6 to 8 months, with no side effects, improved 263 appetite and physical activity [60].”: wrong syntax.
  • Line 297: “Th1 lymphocytes secrete INF-γ, which induces”: wrong syntax

Author Response

REVIEWER 1

The manuscript is a comprehensive review about vitamin intake and tuberculosis clinical outcomes. The research question is important, especially in consideration that the major burden of the disease weighs on countries were undernutrition and malnutrition are endemic. Therefore, in my view, the present work represents a useful tool for researchers and policymakers. However, the authors should address some minor comments:

Response: We thank you very much for the encouraging feedback on our manuscript. We followed your suggestions and believe that now the paper is more usable for the scientific community. Thanks to your suggestions the paper, in our opinion, is notably improved. 

Major comments

  • All article needs some English editing, and paragraph about vitamin B1 (lines 119-133) needs extensive English editing.
  • It may be considered to split section 5 into “Discussion” and a more concise “Conclusion”.

 Response: thank you a lot for your suggestions. A native English speaker revised the paper with a special focus as you suggested on Vitamin B1. On the other hand, we have agreed with all the authors to leave the conclusions somewhat longer because the discussion part is actually already present in every vitamin section. Thank you for your suggestion.

Minor comments

  • Whole manuscript: all microorganisms names should be written in italics, without capital letter on the species name: “ tuberculosis

Response: thank you a lot, we modified as you correctly suggested.

  • Whole manuscript: when multiple references are inserted together, they should be written between the same parenthesis, as follows:[1, 2] or [1, 3-6].

Response: thank you we modified following journal indications

  • Whole manuscript: references should be cited only by first author family name. All given names should be removed.

Response: thank you we modified following journal indications

  • Line 45: “this is just the tip of the iceberg” needs reference

Response: thank you it is only our opinion

  • Line 82: “A second Indonesian trial, found”: wrong syntax.

Response: We corrected the sentence

  • Line 87: “cross-sectional, observational, and supplementation clinical trials”: needs rewording.

Response: we corrected the sentence

  • Line 96-97: study 21 is cited twice in the same sentence.

Response: we corrected it

  • Line 112: the term “consumption” needs definition and reference.

Response: we corrected it

  • Line 127: “TNF-α TNF-” needs correction

Response: we corrected it

  • Line 136: “In a 136 recent paper” which one?

Response: we add reference

  • Line 142: bold characters should be removed

Response: we correct it

  • Lines 141-145: sentence is too long. Also, MTB should be written “ tuberculosis”.

Response: thank you a lot we modified as you suggested

  • Line 147: “Also it is clear that the thiamin biosynthetic pathway can be targeted by 147 drugs” Needs reference.

Response: thank you, we add refereces

  • Lines 148-150: “Studying mycobacterial genes from the perspective of new drug targets is therefore imperative, and pathways leading to the biosynthesis of biotin, thiamine and pyridoxine form interesting study areas.” This sentence is more fitting, potentially, among the conclusions.

Response: thank you for your suggestions

  • Line 154, 164, 274: The first time an acronym is used it should be placed between parentheses.

Response: we corrected it

  • Line 164: “Reactive oxygen species (ROS). [30].” The dot should be removed.

Response: thank you a native English speaker revised the manuscript

  • Line 166: “The pdx1 gene is disrupted, resulting in a B6 auxotrophic MTB mutant.” This sentence needs explanation.

Response: thank you a native English speaker revised the manuscript

  • Line 173: “It acting as a cofactor in acyl CoA carboxylase and pyruvate carboxylase”: wrong syntax.

Response: thank you a native English speaker revised the manuscript

  • Line 176: “as evidenced by genetic research”: reference needed.

Response: thank you we added reference

  • Lines 178 – 179: enzyme names are written both with and without capital letter.

Response: thank you a native English speaker revised the manuscript

  • Lines 184-85: “Consequently, M. Tuberculosis needs biotin production so we can watch at enzymes 184 that synthetize biotin as promising drug targets for new antibiotics” needs rewording.

Response: thank you a native English speaker revised the manuscript

  • Line 192: “ATP-binding cassette-type protein. [37]” wrong syntax. The same for lines 198, 201, 213, 220, 225, 234, 269, 286, 295, 301, 307, 314, 335.

Response: thank you a native English speaker revised the manuscript

  • Lines 234-238: [48] is cited multiple times in the same paragraph.

Response: we modified as better for the paragraph.

  • Line 238: “They compares these results with the study by Kondo e Kanai”: wrong syntax.

Response: thank you a native English speaker revised the manuscript

  • Line 262-264: “Charpy et al. revealed that terminally ill TB patients were given daily 262 high doses (15 g/day) of vitamin C orally for 6 to 8 months, with no side effects, improved 263 appetite and physical activity [60].”: wrong syntax.

Response: thank you a native English speaker revised the manuscript

  • Line 297: “Th1 lymphocytes secrete INF-γ, which induces”: wrong syntax

Response: thank you a native English speaker revised the manuscript

Reviewer 2 Report

In this manuscript authors extensively reviewed the role of vitamins in the treatment and prevention of tuberculosis. Though this is a comprehensively reviewed manuscript, but it has a major shortcoming in English language and scientific writing. For example,

  • Redundancy, first line of Abstract is same as first line of introduction.
  • Lack of continuity in sentences, authors describe latency as “tip of iceberg” but they do not elaborate, why they think so?
  • Grammatical mistakes throughout the manuscript.
  • Manuscript require extensive proof reading for,
  • punctuations,
  • space between the letters or word or punctuations
  • Spellings

I suggest professional editing for this manuscript to be accepted

Materials and method section should be elaborated,

Authors may use a flow chart showing inclusion and exclusion criteria for the relevant articles they used in the study. They should state the total number of articles appeared in the search. How many were sorted? What was the criteria of sorting? and finally, how many are included in the study.

Author Response

Reviewer 2

In this manuscript authors extensively reviewed the role of vitamins in the treatment and prevention of tuberculosis. Though this is a comprehensively reviewed manuscript, but it has a major shortcoming in English language and scientific writing. For example,

  • Redundancy, first line of Abstract is same as first line of introduction.
  • Lack of continuity in sentences, authors describe latency as “tip of iceberg” but they do not elaborate, why they think so?
  • Grammatical mistakes throughout the manuscript.
  • Manuscript require extensive proof reading for,
  • punctuations,
  • space between the letters or word or punctuations
  • Spellings

I suggest professional editing for this manuscript to be accepted.

Response: We thank you very much for the encouraging feedback on our manuscript. Following your suggestions,a native English speaker revised the manuscript  and now we believe that the paper is more usable for the scientific community. Thanks to your suggestions the paper, in our opinion, is notably improved. 

Materials and method section should be elaborated,

Authors may use a flow chart showing inclusion and exclusion criteria for the relevant articles they used in the study. They should state the total number of articles appeared in the search. How many were sorted? What was the criteria of sorting? and finally, how many are included in the study.

Response: Thank you so much for your suggestion. Because this is a narrative review rather than a systematic review, the authors' choice of articles is at their discretion based on a personal criterion of representativeness and relevance, as per research methodology. This is the drawback of conducting a literature review.

What you're asking for is a scoping review or a systematic review, which our article regrettably isn't. As a result, our research uses a simpler methodology, but we hope it will be just as beneficial to the scientific community in understanding how tb interacts with vitamins.

Reviewer 3 Report

In this review, Patti G et al., done a deep search on diverse websites to identify articles discussing the role of Vitamins A, B, C, D and E and Tuberculosis. Authors concluded that the use of multiple micronutrients and vitamins may improve outcome by helping both nutritionally and by interacting directly and/or indirectly with MTB.

This topic is very important in the tuberculosis field, mainly in the search of new supplementary therapies to improve the outcome of the TB patients. From my view’ point some changes in the manuscripts are necessary to improve it.

Mayor comments

There are a lot old references, is possible replace some them by more recent reports?

PLEASE homogenise the use of abbreviations (see minor comments).

Minor comments

  1. Line 44 you indicated that “TB” is the abbreviation to tuberculosis, but after you used “tuberculosis” again in lines 49, 78, 105, 108, 137 (among others), and in the line 54 “tb” or “Tuberculosis” (line 233)
  2. Line 50 you used “vit D” but you did not indicate abbreviation for Vitamin, please use complete or abbreviation but homogenously.
  3. Line 62 you used the abbreviation “VA” to indicate vitamin A, but in line 64, 65, 68, 77 (among others) you used again “vitamin A”
  4. Line 58 you use MTB to indicate Mycobacterium tuberculosis but in lines 74, 144, 145 (among other) you used the abbreviation M. tuberculosis and M. Tuberculosis.
  5. Line 117, why VITAMIN is in capital letter?
  6. Line 120 you used VB (I think this means Vitamin B) please clarify means (maybe in the line 119), some observation to VB5 used in line 154, B7 in the line 173.
  7. Why Vitamin B2 is not abbreviated as another vitamins B?
  8. Line 132, what means MTB tuberculosis? Tuberculosis induced by Mtb? Or you are doing reference to the MTB bacilli (TB means tuberculosis, so you don’t need repeat).
  9. Why FMN is in bold? (line 142).
  10. Indicate what means BMDM in line 158.
  11. Vitamin C also you mix the use of VC and vitamin C, please homogenise.
  12. Line 276 you used “VD” but you did not indicate what means.
  13. Why vitamin E is not abbreviate when the paragraph is started?

Author Response

Reviewer 3

In this review, Patti G et al., done a deep search on diverse websites to identify articles discussing the role of Vitamins A, B, C, D and E and Tuberculosis. Authors concluded that the use of multiple micronutrients and vitamins may improve outcome by helping both nutritionally and by interacting directly and/or indirectly with MTB.

This topic is very important in the tuberculosis field, mainly in the search of new supplementary therapies to improve the outcome of the TB patients. From my view’ point some changes in the manuscripts are necessary to improve it.

Response: We thank you very much for the encouraging feedback on our manuscript. We followed your suggestions and believe that now the paper is more usable for the scientific community. Thanks to your suggestions the paper, in our opinion, is notably improved. 

Mayor comments

There are a lot old references, is possible replace some them by more recent reports?

Response: thank you very much for your suggestion. We have as far as possible done a reference update

PLEASE homogenise the use of abbreviations (see minor comments).

Response: thank you. We modified as your suggestions.

Minor comments

  1. Line 44 you indicated that “TB” is the abbreviation to tuberculosis, but after you used “tuberculosis” again in lines 49, 78, 105, 108, 137 (among others), and in the line 54 “tb” or “Tuberculosis” (line 233)
  2. Line 50 you used “vit D” but you did not indicate abbreviation for Vitamin, please use complete or abbreviation but homogenously.
  3. Line 62 you used the abbreviation “VA” to indicate vitamin A, but in line 64, 65, 68, 77 (among others) you used again “vitamin A”
  4. Line 58 you use MTB to indicate Mycobacterium tuberculosis but in lines 74, 144, 145 (among other) you used the abbreviation M. tuberculosis and M. Tuberculosis.
  5. Line 117, why VITAMIN is in capital letter?
  6. Line 120 you used VB (I think this means Vitamin B) please clarify means (maybe in the line 119), some observation to VB5 used in line 154, B7 in the line 173.
  7. Why Vitamin B2 is not abbreviated as another vitamins B?
  8. Line 132, what means MTB tuberculosis? Tuberculosis induced by Mtb? Or you are doing reference to the MTB bacilli (TB means tuberculosis, so you don’t need repeat).
  9. Why FMN is in bold? (line 142).
  10. Indicate what means BMDM in line 158.
  11. Vitamin C also you mix the use of VC and vitamin C, please homogenise.
  12. Line 276 you used “VD” but you did not indicate what means.
  13. Why vitamin E is not abbreviate when the paragraph is started?

Response: Many thanks for your suggestions. We agree with all of your suggestions and homogenised the use of abbreviations. Also, a native English speaker revised the paper.